# Evaluation of a personalised adherence intervention to improve photoprotection in adults with Xeroderma Pigmentosum (XP): protocol for the trial of XPAND

Jessica Walburn,[1] Sam Norton,[2,3] Robert Sarkany,[4] Kirby Sainsbury,[5] Vera Araújo-Soares,[5] Myfanwy Morgan,[1] Martha Canfield,[2] Lesley Foster,[4] Jakob Heydenreich,[6] Paul McCrone,[7] Adrian Mander,[8] Falko F Sniehotta,[5,9] Hans Christian Wulf,[6] John Weinman[1]

For numbered affiliations see end of article.

**Correspondence to**
Dr Jessica Walburn;
jessica.2.walburn@kcl.ac.uk

## ABSTRACT

**Introduction** Poor adherence to photoprotection for people with xeroderma pigmentosum (XP) can be life-threatening. A randomised controlled trial (RCT) is being conducted to test the efficacy of a personalised adherence intervention (XPAND) to reduce the level of ultraviolet radiation (UVR) reaching the face, by improving photoprotection activities in adults with XP.

**Methods and analysis** A two-armed parallel groups RCT, where we randomised 24 patients with suboptimal adherence to either an intervention group who received XPAND in 2018 or a delayed intervention group who will receive XPAND in 2019. XPAND involves seven sessions, one-to-one with a facilitator, using behaviour change techniques and specially designed materials to target barriers to photoprotection. Following baseline assessment in April 2018 (t0) and intervention, the primary outcome will be measured across 21 consecutive days in June and July 2018 (t1). The primary outcome is the average daily UVR dose to the face (D-to-F), calculated by combining objective UVR exposure at the wrist (measured by a dosimeter) with face photoprotection activities recorded on a daily UVR protection diary. Secondary outcomes include average daily UVR D-to-F across 21 days in August (t2); psychosocial process variables measured by daily questions (t0, t1, t2) and self-report questionnaires (t0, t1, t2, December 2018 (t3)). Intervention cost-utility is assessed by service use and personal cost questionnaires (t0, t3). The delayed intervention control arm participants will complete three further assessments in April 2019 (t4) and June–July 2019 (t5), and December 2019 (t6) with dosimetry and UVR protection diary completed for 21 days at t4 and t5. A process evaluation will be conducted using mixed methods.

**Ethics and dissemination** Ethical approval has been received from West London & GTAC REC 17/LO/2110. Results will be disseminated in peer-reviewed journals and at conferences. This study tests a novel intervention, which, if successful, will be integrated into routine care.

**Trial registration number** NCT03445052; Pre-results.

## INTRODUCTION

Xeroderma pigmentosum (XP) is a very rare genetic condition, where the body is unable

### Strengths and limitations of this study

► To the best of our knowledge, this is the first randomised controlled trial to evaluate an adherence intervention designed to improve photoprotection in people diagnosed with xeroderma pigmentosum.
► We have a primary outcome which is clinically relevant.
► We control for seasonal differences in environmental ultraviolet radiation exposure by comparing between groups across the same weeks.
► We include a process evaluation to understand participant views of XPAND which will be important when integrating the intervention into current clinical care.
► Recruitment and retention of participants may be difficult in this extremely rare disease.

to repair ultraviolet radiation (UVR)-induced damage to DNA caused by daylight.[1] Any UVR exposure dramatically increases the chances of developing skin and eye cancers (eg, rates of non-melanoma skin cancer increase 10 000-fold).[2] Photoprotection is *recommended* for the healthy population and those taking medications which may cause photosensitivity. However, it is *essential* for people with XP. There is currently no cure for this disease and the only way to minimise and delay the cancers is meticulous and absolute photoprotection every day. Photoprotection involves a complex set of behaviours (ie, application of broad-spectrum SPF50 sunscreen, wearing long sleeves and long trousers or skirts, wide brimmed hat, glasses, face scarf or buff and avoiding UVR exposure by staying indoors). The best way of protecting the face is by wearing a face visor (a legionnaire style hat with a UVR-protective transparent film at the front). Analysis of

self-reported photoprotection used by 21 adults with XP in the UK revealed that over half were using 'very poor' or no photoprotection for at least 20% of all outdoor time.[3] This poor photoprotection has life-threatening consequences.

In common with other rare conditions,[4] little is known about the psychological and behavioural patient-related factors that have the potential to improve health outcomes in patients with XP. In recognition of this deficit, the National Institute of Health Research (NIHR) agreed to fund a novel programme of research to investigate the psychosocial factors associated with non-adherence to photoprotection in XP (RP-PG-1212–20009: Developing a psychological intervention to improve ultraviolet protection and clinical outcomes in XP). The aims of the grant were twofold: to identify the drivers of non-adherence using complex mixed-methods research (qualitative interviews; cross-sectional survey; n-of-1; diary-dosimeter measures)[5] (known as 'Phase One') and to use this knowledge to design and test an intervention to improve adherence ('Phase Two'). We identified 17 modifiable psychosocial drivers of non-adherence to photoprotection activities from Phase One. These drivers or barriers to optimal photoprotection relate to factors influencing the motivation to protect (eg, doubts about perceived necessity of photoprotection) and volitional factors limiting the enactment of photoprotection activities, even if motivation is high (eg, lack of habit). Findings from the mixed-methods research studies are published separately.[3 6 7]

We aimed to systematically develop an adherence intervention to improve photoprotection in XP. There is growing support for interventions focused on changing patient beliefs about their illness and treatment, translating into improvements in adherence across a range of chronic conditions.[8–10] This suggests that addressing unhelpful illness and treatment beliefs may be an important treatment target for photoprotection in XP. Furthermore, a recent systematic review found evidence in support of the effectiveness of psychosocial interventions to improve adherence to photoprotective activities among non-XP individuals at elevated risk for melanoma (due to personal or family history[11]).

We designed a personalised intervention titled, 'XPAND: Enhancing XP Photoprotection Activities – New Directions', using intervention mapping.[12] Intervention mapping starts by breaking the desired behaviour and determinants of behaviour into constituent parts, which are mapped to theory and translated to intervention components. Relevant theories to guide the intervention design included the Necessity and Concerns Framework,[13] Common Sense Model of Illness Regulation[14] and the Theoretical Domains Framework (not a theory itself but it combines the components of social cognition models used to explain behaviour).[15] A full description of the development of the intervention, the process of personalisation and the final intervention product will be reported separately.

This paper describes the protocol for a randomised controlled trial (RCT) to test the efficacy of XPAND to lower the dose of UVR reaching the face, by improving adherence to photoprotection. This can be achieved by minimising overall UVR exposure (ie, time spent outside) and by increasing level of photoprotection worn or applied when outside. The primary objective is to reduce the average daily UVR dose to the face (D-to-F) across 21 days in June to July (2018), immediately after the delivery of the main intervention. Secondary objectives are: to maintain reductions in average daily UVR D-to-F across 21 days in August 2018 (after a booster session has been delivered to the intervention group); to increase and then maintain daily ratings of mood, self-efficacy, goal priority, automaticity and photoprotection activities across the two 21-day periods. Tertiary objectives are to explore intervention-related changes from baseline. A qualitative process evaluation will investigate the acceptability, feasibility and change mechanisms from the perspective of the participant. We will conduct a cost-utility analysis, which will indicate whether implementation into clinical care is economically viable.

## METHODS AND ANALYSIS
### Participants and recruitment
People diagnosed with XP and registered at the National XP Service at Guy's and St Thomas' NHS Foundation Trust will be recruited into the trial. Eligible patients will be sent an invitation letter, followed by a telephone call from the research nurse, and those interested will be sent the study information leaflet. Following a second call, if they wish to participate, they will give written consent at a home visit. Patients will be reminded that participation in the study is voluntary and that they may withdraw at any time. No payment will be given for participation. Screening and recruitment will take place between February and March 2018.

### Inclusion criteria
1. A confirmed diagnosis of XP (reduced DNA repair activity in DNA fibroblast assay, in a clinical context compatible with a diagnosis of XP and with confirmation by finding pathogenic mutations in an XP gene on gene sequencing)
2. Aged ≥16 years
3. Suboptimal adherence to photoprotection when outdoors, as identified by the XP clinical team from data held in medical notes, or by the research team from data collected during the Phase One studies:
   a. Score <20 on our Adherence to Photoprotection scale (where a total score of 25 indicates optimal photoprotection)[16]
   b. Using photoprotective clothing combinations for the face that have been assessed by the clinical team as anything other than 'excellent' or 'very good' anytime outdoors,[3] recorded on the daily UVR protection diary

c. Having a 'resistant' or 'integrated' mode of adjustment to photoprotection, as identified in the qualitative analysis[7]

## Exclusion criteria

1. Diagnosed with cognitive impairment (XP or non-XP related) due to potential impact on the efficacy of the intervention and on the participants' experiences of taking part.
2. Non-fluent in English (to enable in-depth discussion with the intervention facilitators)
3. Diagnosed with current clinical depression or anxiety, as detailed in medical notes, or confirmed after completion of Hospital Anxiety and Depression Scale (HADS)[17] at the baseline home visit

## Study design and flowchart

The XPAND trial is a phase II two-armed parallel group RCT, with a delayed intervention control arm. Participants are randomised to either the intervention group who receive the XPAND intervention in addition to routine care between April and June 2018 or the control group who receive routine care in 2018 and then XPAND between April and June 2019. The delayed intervention control group is included to maximise information collection about the intervention due to the extremely rare nature of XP. For example, achievable sample sizes limit the ability to test between-group differences for the psychosocial outcomes with acceptable power and the delayed intervention group allows for intervention-related changes from baseline testing, which is less robust but still useful.

Patient flow through the study is described in figure 1. The baseline assessment takes place in April 2018 (t0). Post-randomisation assessments take place after completion of the intervention in June to July 2018 (t1), after a booster session in August 2018 (t2) and after a long-term follow-up in December 2018 (t3). All participants use a wrist-worn UVR dosimeter continuously from t0 to t2 and complete daily UVR protection diaries for 21 consecutive days at each assessment time point. At each assessment, participants additionally complete patient-reported outcomes. The t3 follow-up does not involve dosimetry

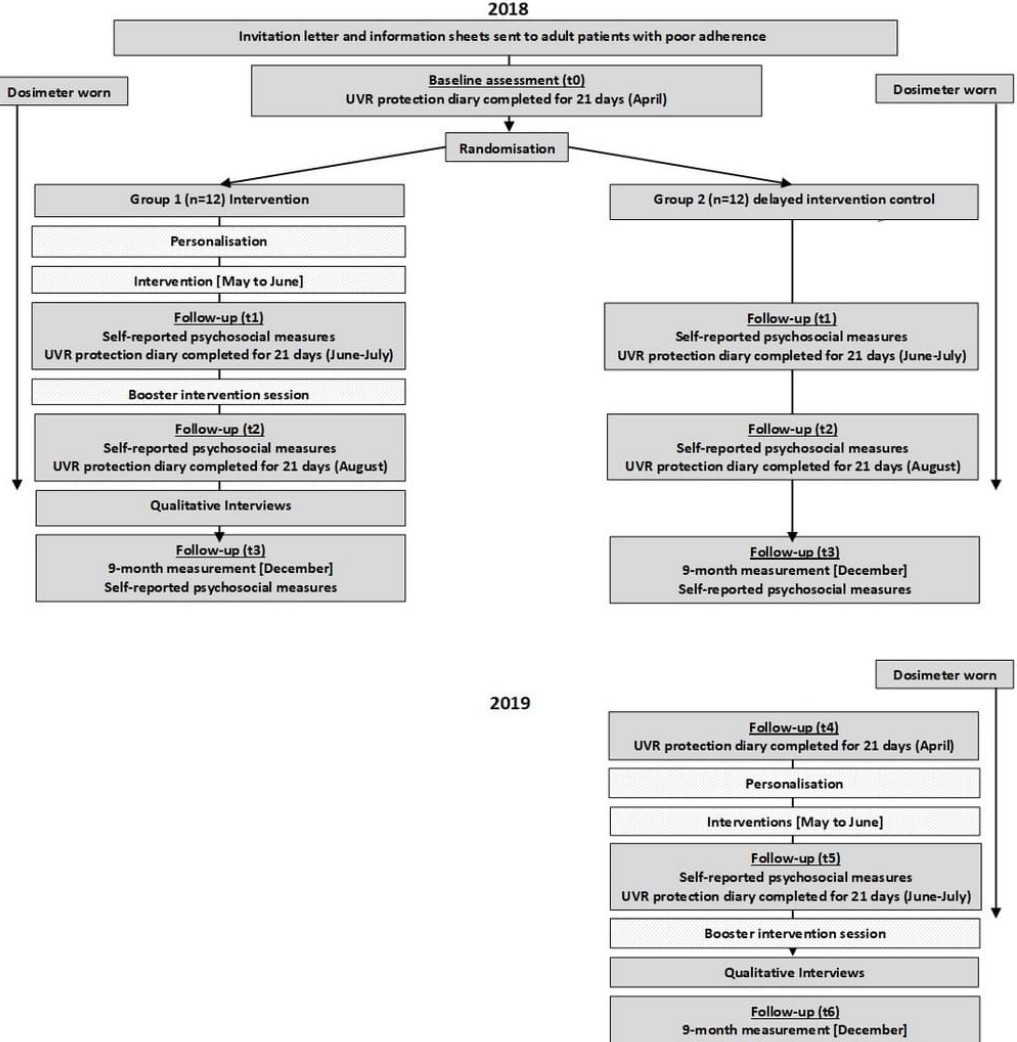

**Figure 1** Progression of participants through the study.

or daily UVR protection diary. In addition, the delayed intervention control arm only completes three further assessments in April 2019 (t4) and June–July 2019 (t5), and December 2019 (t6) with dosimetry and daily UVR protection diary completed for 21 days at t4 and t5.

A nested qualitative study examining acceptability and mechanisms of change will be conducted in the intervention group after the August assessment period is completed.

The study is funded by the NIHR Programme Grants for Applied Research scheme (RP-PG-1212–20009) and received ethical approval.

### Randomisation

Participants will be randomised, in blocks using an equal allocation ratio, to receive XPAND immediately (2018) or to the delayed group stratified by burning type to attempt to balance those with an extreme (ie, scoring between 1 and 3 on the sunburn severity score[18]) versus normal burning response. Participants who are in the same family will be randomised as a cluster to the same group to avoid contamination. Since all participants will be recruited at the point of randomisation, the trial statistician (SN) will generate a random allocation sequence for all participants together, using a computer programme with fixed block sizes of 4, to ensure equal allocation to both groups. To protect the integrity of the randomisation, participants in the immediate intervention group will be asked not to reveal their allocation to those outside their immediate family. Group allocation will be concealed from the XP clinical team who are not part of the research team (excluding the PI) to avoid inadvertent changes to the standard care of these participants during the trial (eg, greater/lesser discussion of adherence during routine clinical appointments). Moreover, as both the control and intervention groups might receive opportunistic encouragement from the clinical team to protect, this would not adversely affect the trial outcome.

### Primary outcome

The primary outcome is the average daily UVR D-to-F (standard erythemal dose (SED)), across 21 consecutive days between June and July 2018 (t1), following completion of six out of seven intervention sessions by the intervention group. UVR D-to-F is estimated by combining data from a UVR dosimeter worn on the wrist (SunSaver 3, Bispebjerg Hospital, Copenhagen, Denmark)[19] and the patient's photoprotective activities, self-reported using the daily UVR protection diary. The dosimeter was set to measure UVR levels every 5 s and record the average every 5 min, which has been validated as providing reliable measurements of total UVR exposure over the course of a day.[20] The combination of the daily UVR protection diary and dosimeter to estimate UVR D-to-F has been used in our earlier research with this population (Phase One) and demonstrated to function sufficiently well to be the primary outcome in this trial. The paper-based daily UVR protection diary allows participants to record periods of

time spent outside (in 15 min intervals) and the timing of any specific photoprotection activities while outside (ie, wearing a face visor, hat, hoodie worn-up, glasses, scarf or face buff or applying sunscreen and lip block) by drawing a line on a grid (see online supplementary file 1). To avoid errors in recording, each participant will receive training on how to complete the diary. UVR D-to-F will be calculated using the dose recorded at the wrist weighted by the level of protection provided by the combination facial photoprotection behaviours used.[5]

### Secondary outcomes

The following outcomes relate to between-group comparisons in 2018 and are recorded on the daily UVR protection diary. UVR-related outcomes also use data from the dosimeter.

a. Average daily UVR D-to-F across 21 consecutive days in August 2018 (t2). This is to investigate the stability of any improvement identified in the primary outcome period (June to July 2018), as it will be 4–6 weeks after the main intervention and follows session 7 (the booster session).

b. Average daily total UVR exposure during each of the 21 day periods (t1, t2) and the average daily time outside across all daylight hours and restricted to 11:00–15:00, when UVR levels are at their highest. These data will be gathered from the dosimeter and the UVR protection diary.

c. Average daily proportion of time outside across daylight hours and between 11:00 and 15:00 with 'very good' or 'excellent' face photoprotection during each of the 21 day periods (t1, t2). The categorisations relate to the Daily Photoprotection Scale (DPS)[3] developed for the Phase One n-of-1 study in consultation with the UK XP clinical team. Facial photoprotection activities (excluding sunscreen) are ranked according to their relative level of photoprotection. The following combinations are classed as 'very good photoprotection': hat, glasses and scarf/buff; hat, glasses, scarf/buff and hoodie. 'Excellent photoprotection' is wearing a face visor, which completely covers the face.

d. Average daily proportion of time outside across daylight hours with sunscreen protection, defined as when sunscreen was applied within the previous 3 hours, during each of the 21 day periods (t1, t2).

e. Average daily number of times sunscreen is applied irrespective of time outside during each of the 21 day periods (t1, t2).

f. Average daily measures of mood, automaticity of photoprotection activities, prioritisation of photoprotection compared with other priorities and level of self-efficacy to manage barriers during the two follow-up periods (t1, t2) will be assessed. The single items have been adapted from ecological momentary assessment questions used in our n-of-1 study[3] (eg, How much do you agree that UVR protection of your face today was something you did automatically without thinking (0=strongly disagree to 10=strongly agree)) and are

**Table 1** Summary of self-reported measures schedule

| | Intervention and control groups 2018 | | | | Delayed intervention group only 2019 | | |
| --- | --- | --- | --- | --- | --- | --- | --- |
| | Baseline (t0) (April) | Follow-up (t1) (June–July) | Follow-up(t2) (August) | Follow-up (t3) (December) | Follow-up (t4) (April) | Follow-up (t5) (June– July) | Follow-up (t6) (December) |
| UVR Protection Diary | ✓ | ✓ | ✓ | | ✓ | ✓ | |
| EuroQol five dimensions questionnaire (EQ-5D) | ✓ | ✓ | ✓ | ✓ | ✓ | ✓ | ✓ |
| Short-form Warwick-Edinburgh Mental Well-Being Scale (SWEMWBS) | ✓ | ✓ | ✓ | ✓ | ✓ | ✓ | ✓ |
| Self-Report Behavioural Automaticity Index' (SRBAI) | ✓ | ✓ | ✓ | ✓ | ✓ | ✓ | ✓ |
| Photoprotection Self-efficacy Questionnaire (PhotoSEQ) | ✓ | ✓ | ✓ | ✓ | ✓ | ✓ | ✓ |
| Brief Photoprotection Adherence Questionnaire (BPAQ) | ✓ | ✓ | ✓ | ✓ | ✓ | ✓ | ✓ |
| Service Use Questionnaire (SUQ) | ✓ | | | ✓ | ✓ | | ✓ |
| Feedback questionnaire (intervention group only) | | ✓ | | | | ✓ | |

included on the UVR protection diary (see online supplementary file 1).

### Tertiary outcomes

The following outcomes are assessed once, at the start of each 21 day period (t0, t1, t2, t3). Health-related quality of life (HRQoL), psychological well-being, automaticity of photoprotection activities, self-efficacy in the context of barriers and self-rated photoprotection adherence will be measured using self-report questionnaires. Data will be combined across t1 and t2 to assess proximal impact of the intervention and t3 will be used to assess stability of impact.

See table 1 for the schedule of self-reported measures.

a.  HRQoL will be assessed using the EQ-5D-5L[21]

b.  Emotional well-being will be measured by the Short-form Warwick Edinburgh Mental Well-Being Scale (SWEMWBS)[22]

c.  Automaticity of photoprotection activities will be assessed using the 4-item Self-Report Behavioural Automaticity Index (SRBAI),[23] adapted to photoprotection. It is a validated subscale of the Self-Report Habit Index[24] which focuses on the automaticity of behaviour. It asks respondents to rate the extent to which they agree with each statement on a seven-point Likert-type scale (eg, UVR photoprotection is something I do without thinking (1=strongly disagree to 7=strongly agree)). The average of responses gives a score between 1 and 7, with higher scores indicating greater automaticity.

d.  Self-efficacy to photoprotect will be measured using a 21-item scale (Photoprotection Self-Efficacy Questionnaire (PhotoSEQ)) developed for this study, as no validated questionnaire specific to photoprotection

activities in the presence of barriers, as recommended by Bandura,[25] was identified (see online supplementary file 2). Three items ask the respondent to rate their level of confidence that they can carry out a type of photoprotection activity (eg, shifting timing and/or duration of outdoor activity, photoprotection using clothing, correctly apply sunscreen) on a 10-point scale (0=Not at all – 10=very confident); two items ask separately about confidence to wear photoprotective clothing and apply sunscreen in the presence of nine different barriers in the following 4 weeks (eg, How confident are you that you can photoprotect even if/ when: unexpected things get in the way). Two subscales are calculated, an average self-efficacy score for sunscreen and similarly for photoprotective clothing and shifting time/duration of activity. Higher scores indicate greater self-efficacy.

e.  A secondary brief measure of photoprotection activities was developed to assess photoprotection to indicate whether improvements in photoprotection are maintained, when participants will not be completing the daily UVR protection diary at t3 (see online supplementary file 3.). To allow comparison with the diary, it will be completed at all follow-up points. The Brief Photoprotection Adherence Questionnaire (BPAQ) has five items and assesses duration of time outdoors, and photoprotection used when outdoors during the previous 7 days. Three items assess different ways of protecting outdoors: how often respondents wore protective clothing (eg, When you went outside, how often did you protect your face against UVR using protective clothing? (0=never to 10=all the time)); the number of days sunscreen was applied in the morning (0–7 days);

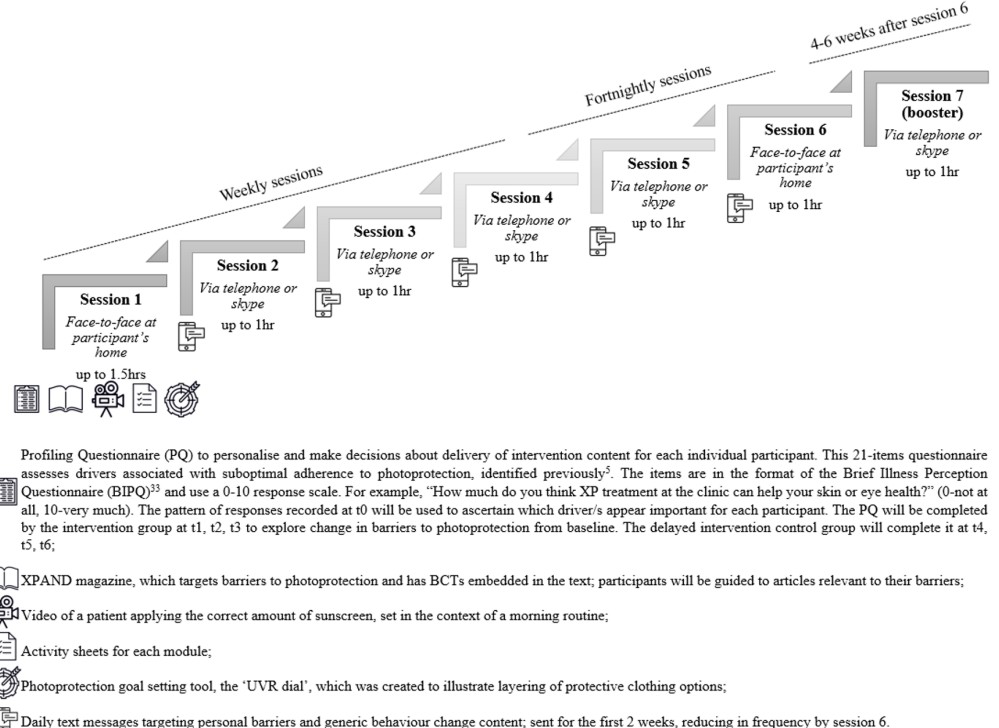

**Figure 2** Structure of the XPAND intervention.

and how often sunscreen was reapplied when outdoors for longer periods (0=not at all to 10=always). Two items ask respondents to estimate average duration of time outdoors each day (Never, <30 min, 31 min–1 hour, 2 hours, upwards in hourly intervals to >8 hours) and average duration of time outdoors each day between 11:00 and 15:00 when environmental UVR levels are highest. Each item will be analysed separately.

Further exploratory analyses will examine change from baseline incorporating the delayed intervention group allowing for higher power to assess intervention-related changes. These will be undertaken unblinded to group allocation after main efficacy analyses are complete.

### The XPAND intervention

XPAND is composed of a combination of one-to-one sessions and materials purposely designed to target barriers to photoprotection. It was designed to be delivered by a healthcare professional (HCP) without specialist psychological training (eg, clinical nurse specialist). In the trial context, facilitators will be two psychologists and a research nurse.

### One-to-one sessions

Facilitators will deliver 7x one-to-one sessions to each participant. Further information on the structure of the intervention is available in figure 2. Each session will include a combination of personalised and generic content. The intervention content and techniques were systematically developed and selected using intervention mapping.[12] Core strategies delivered to all participants will include those to increase self-efficacy, self-regulatory

skills and the automaticity of photoprotection. In addition, personalised modules will be delivered, as needed, selected initially based on data collected in Phase One studies and a profiling questionnaire, and iteratively as additional barriers emerge during the sessions. The facilitators will use a communication style consistent with motivational interviewing.[26] They will be guided by a manual, including theory-based behaviour change methods mapped to recognised taxonomies,[12 27] and specific techniques from other therapeutic approaches (eg, Acceptance and Commitment Therapy).[28]

### Intervention materials

Patient-facing materials have been developed and will be provided to participants in session one and used throughout the intervention to facilitate discussion, provide real-life examples and reinforce the concepts and skills discussed during the sessions. These are summarised in figure 2.

### Sample size calculations

The target sample size for the study is a total of 24 patients (ie, 12 per group), which was based on the ability to detect an average daily UVR D-to-F difference of 0.10 SED between the groups during the June to July 2018 assessment period. This difference was considered clinically meaningful and potentially achievable due to the personalised nature of the intervention. From our Phase One study of UVR exposure, the average daily UVR D-to-F for people diagnosed with XP was 0.27 SED (SD=0.14), with a within-person correlation of $r_{ICC}=0.31$. Thus, the anticipated difference relates to an effect size of d=0.73,

or just over a one-third reduction in average daily UVR D-to-F. Adjusting for the design effect due to the dependence of assessments within individuals, a sample size of 20 patients with 21 daily observations has 80% power to detect a reduction of 0.10 SED in average daily UVR D-to-F at the 5% significance level. Accounting for attrition of 20%, based on our previous research, the target sample size was set at 24.

### Statistical analyses

The main efficacy analyses will be conducted by the trial statistician (SN) following a prespecified analysis plan and blind to group allocation. The analysis will follow the intention to treat principle with individuals analysed within the groups to which they were randomised irrespective of whether they received or persisted with the intervention.

The treatment effect on the primary outcome of daily UVR D-to-F over 21 days between June and July 2018 and secondary outcome of daily UVR D-to-F over 21 days in August will be estimated simultaneously using a linear mixed model. Given the skewed distribution of UVR D-to-F, a logarithmic transformation will be applied and robust SE will be estimated. A random intercept will account for the repeated assessments of UVR D-to-F within individuals across all 42 days, with an autoregressive error structure to account for correlation between assessments across successive days. In addition to the dummy-coded group variable, other covariates in the model include a dummy-coded variable for the assessment period (June–July vs August), an assessment period by group interaction, the patients baseline average daily UVR, an indictor variable for the patients propensity to burn and the background level of UVR recorded by the observatory closest to the patients' house on the day. Data will be summarised as the empirical Bayes estimate of the average daily UVR D-to-F in SEDs for each assessment period for each group.

This approach allows for missing daily data within each individual under the assumption that data are missing at random. We will run sensitivity analyses to explore the influence of missing data on the effect estimates, so as to allow for a consideration of the likely plausible range for the treatment effect under a number of missing not at random assumptions. For average daily assessments, where these are not available for the full 21 days, this will involve imputing missing values under a range of conservative assumptions using the average daily values of the group to which the individual is assigned using a pattern mixture model approach (eg, average daily level +0.0 SED to 0.5 SED).

Treatment effects for average daily levels of mood, self-efficacy, goal priority, automaticity and time outside will be analysed for both assessment periods simultaneously using mixed-effects as described above. Treatment effects for adherence, self-efficacy, automaticity, HRQoL and psychological well-being taken at the start of the June and August assessment periods will also be estimated using linear mixed models but will specify an unconstrained error structure. It is not anticipated that transformation will be required to account for skew in any of these outcomes; however, robust SE will be estimated. Covariates included in the analysis will be the same as above except that baseline levels of the outcomes will be included in place of baseline average daily UVR D-to-F. Scale scores will be calculated using proration to account for missing items within the scale where there is at most one-third of items missing. For example, where scale includes six items, at least four items must have been completed. Otherwise, the scale score will be set to missing for that individual.

Further exploratory analysis using dynamic regression models will evaluate changes in both level and variability in daily UVR exposure over time between March and September, accounting for treatment using a regression discontinuity approach. This is possible as the dosimeter will be worn for the entire period. However, due to patient burden and since the UVR protection diary is only completed for 3-week intervals, it will also be necessary to calculate UVR D-to-F using average daily photoprotection during the pre and post diary intervals. This differs from the main analysis where UVR D-to-F is calculated directly based on photoprotection activities recorded on the diary.

### Process evaluation

A mixed-methods process evaluation, using qualitative interviews and self-report measures, will explore the acceptability of the intervention, changes in photoprotection activities and psychosocial mechanisms of change. A brief intervention feedback questionnaire, completed immediately after session 6, will be used to record more proximal perceptions of the main content of intervention. Responses to this questionnaire and the profiling questionnaire (see figure 2) will be used to guide prompts during these interviews and give insight into psychosocial mechanisms of change.

The feedback questionnaire is adapted from one used to assess a fat reduction intervention.[29] It has five items assessing overall perceptions of the programme and its components, the impact on photoprotection activities and whether psychosocial variables have changed as a result of the intervention. The questions assess the extent to which respondents agree with statements about the intervention (eg, 'Overall the programme was interesting' (1=strongly disagree to 5=completely agree)) (see online supplementary file 4).

In-depth qualitative interviews will be conducted by a research nurse and health psychologist who were involved in the design and delivery of the intervention. They will not interview participants to whom they delivered the intervention. Interviews will be based on a topic guide and will explore participants' views of and experiences with XPAND, including more practical delivery-related aspects, such as the optimal number of sessions, preference for telephone or skype sessions and the value of any further booster sessions. All interviews will be

audio-recorded, transcribed and coded using thematic analysis[30 31] in NVivo V.10.

### Fidelity assurance

A fidelity assessment will be undertaken to examine the extent to which facilitators delivered the key components of the intervention, specified in the intervention manual, during the face-to-face sessions. Independent researchers will apply a fidelity checklist to audio recordings of all session 1 s and session 6 s, as well as and a random selection of subsequent sessions.

### Evaluation of the cost-utility of XPAND

A decision model will be used to assess the cost-effectiveness of the intervention. Decision models allow outcomes and costs associated with alternative care process to be investigated via simulations. Model structures simplify the care process such that specific aspects can be focused on. Advantages of these models are that they can be adapted to reflect the outcomes and costs that occur in a variety of settings (and therefore aid generalisability); they allow evidence to be generated in a time and cost-efficient way; and they enable interventions to be evaluated that may be precluded using trial methods. The structure of the model to be used here will be developed by the health economists in collaboration with the clinical researchers, and will consist of health states that patients may be in over time. Transition between the states will be informed by literature on the progression of the condition, expert opinion and data collected through the trial. The impact of the intervention on these transitions will be determined through the clinical trial. A health service perspective will be used followed by a societal perspective, which will incorporate broader costs.

Service use will be measured via completion of a Client Service Receipt Inventory (CSRI).[32] This retrospectively assesses use of primary and secondary healthcare services (including surgical interventions), social care, tests/investigations and aids and adaptations (see online supplementary file 5). The cost of these inputs will be calculated by combining the service use data with appropriate unit cost information. Other impacts of XP include additional financial costs for the patient (eg, sunscreen); and time lost from work/education by patients (eg, to attend appointments, receive treatment, manage symptoms). Costs of these effects will be calculated using average wage rates and information on returns to education. Costs will be combined with quality-adjusted life years (QALYs) derived from the EQ-5D-5L. The estimated cost of the intervention will be based on staff time required to deliver it, plus additional training and materials.

### Trial management

An independent Trial Steering Committee (TSC) attended by the research team, two independent researchers and one patient and public involvement (PPI) panel member will meet every 3 months to provide oversight of the trial. Trial data are collected after each measurement period by a member of the research team not delivering the intervention to that individual. Data are managed and entered by the research team, not the statisticians analysing the data. Adverse events are defined as events that occur during participation in the trial. All will be recorded (death; life threatening; hospitalisation; self-harm; attendance at A&E; distress (intervention group)) and reported to the TSC, who will make a judgement on the link to the trial and recommend modifications or discontinuing trial as appropriate.

### Patient and public involvement

PPI has been integral to every step in the development of this protocol. The PPI panel advised on the study design, particularly related to participant burden of completion of the daily UVR protection diary, which informed the decision to limit the follow-up of the delayed intervention control group to a single period of 21 days (June–July 2019, t5). In addition, the panel reviewed all the XPAND intervention participant-facing materials which helped ensure they were appropriate and acceptable. The PPI panel were not involved in the recruitment process. A summary of research findings will be sent to all participants and published papers will be made available.

### DISCUSSION

To the best of our knowledge, this is the first RCT to test a behaviour change intervention to improve adherence to photoprotection in adults with XP; an ambitious undertaking in a rare disease. It uses a robust methodology, both in the systematic development of the intervention and in the use of a clinically relevant novel measurement approach to estimate dose of UVR reaching the face. The inclusion of a process evaluation will provide insight into the workings of the intervention and the mechanism underlying our trial outcomes, as well as information about the acceptability of XPAND, a key consideration for the implementation of the intervention into routine clinical care. We will also have estimates of the cost-effectiveness of the intervention. Furthermore, over and above the obvious clinical advantages of decreasing the risk of morbidity and mortality associated with skin cancer for the UK XP population, if the intervention is efficacious, it will be generalisable to international populations of adults with XP, significant for other conditions requiring photoprotection (eg, Systemic Lupus Erythematosus) and relevant for healthy populations.

### ETHICS AND DISSEMINATION

This research has been approved. The findings will be published in peer-reviewed journals and presented at national and international scientific conferences.

### Trial status

Recruitment completed. Patient involvement in the study will conclude in December 2019.

**Author affiliations**
[1]School of Cancer and Pharmaceutical Sciences, Faculty of Life Sciences and Medicine, King's College London, London, UK
[2]Health Psychology Section, Institute of Psychiatry, Psychology and Neuroscience, King's College London, London, UK
[3]Department of Inflammation Biology, Faculty of Life Sciences and Medicine, King's College London, London, UK
[4]National Xeroderma Pigmentosum Service, Guy's and St Thomas' NHS Foundation Trust, London, UK
[5]Faculty of Medical Sciences, Institute of Health and Society, Newcastle University, Newcastle upon Tyne, UK
[6]Department of Dermatology, Bispebjerg Hospital, Copenhagen, Denmark
[7]Health Service & Population Research Department, Institute of Psychiatry, Psychology & Neuroscience, King's College London, UK
[8]MRC Biostatistics Unit, School of Clinical Medicine, Cambridge Institute of Public Health, University of Cambridge, Cambridge, UK
[9]Institute of Health & Society, Faculty of Medical Sciences, Newcastle University, Newcastle upon Tyne, UK

**Acknowledgements** We would like to acknowledge the valuable role of the XP national clinical team (Hiva Fassihi, Tanya Henshaw, Alan Lehmann, Sally Turner) and the PPI panel (Cathy Coleman, Ben Fowler, Ros Tobin, Sandra Webb) in assisting with development of this protocol. In addition, we would like to acknowledge the support of the NIHR team (Hothan Esmael, Jonathan Gabriel, Katherine Horner, Thomas Hutchinson) and Robert Pleass (Dermatology Research Manager Guy's and St. Thomas' NHS Foundation Trust). Many thanks to Federica Picariello for commenting on the final draft.

**Contributors** JWa drafted the manuscript with assistance from KS and LF. VAS, MC, JH, PM, AM, MM, SN, KS, RS, FFS, JW and HCW reviewed and commented on the manuscript. All authors were involved in study design. JWa, LF and KS are involved in intervention delivery and data collection.

**Funding** This study/project is funded by the National Institute for Health Research (NIHR) (Programme Grant for Applied Research Scheme (RP-PG-1212 20009)). The trial sponsor is Guy's and St. Thomas' NHS Foundation Trust.

**Disclaimer** The views expressed are those of the author(s) and not necessarily those of the NIHR or the Department of Health and Social Care.

**Competing interests** None declared.

**Patient consent for publication** Not required.

**Ethics approval** West London & GTAC Research Ethics Committee 17/LO/2110

**Provenance and peer review** Not commissioned; externally peer reviewed.

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
