## [Reviewer comments · BMJ Open]

ARTICLE DETAILS

TITLE (PROVISIONAL)	Evaluation of a personalised adherence intervention to improve photoprotection in adults with Xeroderma Pigmentosum (XP): Protocol for the trial of XPAND
AUTHORS	Walburn, Jessica; Norton, Sam; Sarkany, Robert; Sainsbury, Kirby; Araújo-Soares, Vera; Morgan, Myfanwy; Canfield, Martha; Foster, Lesley; Heydenreich, Jakob; McCrone, Paul; Mander, Adrian; Sniehotta, Falko; Wulf, Hans Christian; Weinman, John

VERSION 1 – REVIEW

REVIEWER	Kenneth Kraemer and Deborah Tamura National Cancer Institute, USA
REVIEW RETURNED	31-Jan-2019

GENERAL COMMENTS	Evaluation of a personalised adherence intervention to improve photoprotection in adults with Xeroderma Pigmentosum (XP): Protocol for the trial of XPAND. This is a randomized controlled trial to test the efficacy of a personalized adherence intervention (XPAND) to reduce the level of ultraviolet radiation (UVR) reaching the face. The study intervention aims to improve photoprotection activities in adults with XP. The study is supported by the National Institute of Research and has appropriate monitoring through a steering committee. The study has also met all ethical requirements of the regulatory bodies for consent to participate and voluntary withdrawal from the study. This is a very ambitious protocol and the study is certainly warranted as XP adults can be the most resistant to adopting UV protection. The authors/protocol developers have previously documented that UV protection among this group of XP adults is very poor leading to costly surgical interventions, facial disfigurement and psychological distress. In addition, validated educational and social/psychological support and intervention methodologies are essentially lacking in this population of patients. I look forward to the outcome. I have several comments about the study. 1. Methodology: a. This is a very select sample and small size (24 patients); randomization will make the intervention sample size even smaller; it may limit eventual generalization to a larger population. b. Some XP patients (for example those in complementation groups A or D) are much more photosensitive than others (for example XP complementation groups C or XP variant). They indicate that the patients were stratified by burning type. A reference should be provided as to “burning type”. This stratification will further reduce the sample sizes. Most patients who have extreme burning type will probably practice better sun
---

	protection than those who do not have extreme burning because they receive immediate negative feedback by being burned. c. Are you taking into account whether the XP patient had a skin cancer removed in the past? This might be a good motivator toward better sun protection. d. Exclusion criteria include “Diagnosed with cognitive impairment”. This is too vague and should be made more specific. For example, some XP patients with mild cognitive impairment may be more amenable to the education than some XP patients with no impairment. e. Has it been established that the amount of UV reaching the dosimeter on a participant’s wrist modified by the patient’s report is actually related to the amount of UV reaching the face? Could the reliability of the reports vary from one patient to another? I do not see that clearly stated in this paper, although some older papers are cited. The readings from this dosimeter are closely associated with the effectiveness of the intervention and the primary study outcome. The comparability of the readings among different individuals should be demonstrated. f. The Photoprotection Self Efficacy Questionnaire and The Brief Photoprotection Adherence Questionnaire are interesting tools that could be used in other clinics and for other photosensitive conditions. Since the tools haven’t been validated, the participants responses may not adequately answer the research questions. It may be worth doing separate validation of these tools for use in the future. g. How much time will be required for each patient to fill out all of the questions required? Is this time reasonable? Is it reasonable to expect each patient to carry out so many daily observations? Adults who have demanding daily activities (such as jobs) may not be as reliable or able to fill out as much of the materials as others who do not have these responsibilities. On the other hand, repeated focusing on sun protection by thinking about filling out the forms may reinforce the message. 2. Although the clinic personal have been ‘blinded’ to the randomization of patients, there may be ‘bleed over’ of education, support and guidance for the patients who are perceived as not adequately photo-protecting. Most health care personal who work with XP patients, will try to be proactive with all patients who are not photo-protecting. 3. Please remove references to being “first” or “nobody has attempted”. This is scientifically irrelevant and cannot be proven. 4. Please reconcile verb tenses in lines 52 to 60 on page 7 and 3 to 11 on page eight. One area is future tense and the other past tense.
--	---

REVIEWER	Eszter Baltas University of Szeged, Department of Dermatology and Allergology, Hungary
REVIEW RETURNED	04-Feb-2019

GENERAL COMMENTS	The study protocol is very well designed and written.
---

REVIEWER	Shinich Moriwaki Osaka Medical College, Japan
REVIEW RETURNED	08-Feb-2019

GENERAL COMMENTS	The authors proposed the protocol for a RCT to test the efficacy of XPAND to lower the dose of UVR reaching the face by improving adherence to photoprotection. This is the first trial, which may be useful to change a XP patients' behavior about UV protection or a policy of UV care by patients' parents in the future. Study design is good and questionnaires are appropriate to evaluate the outcomes. XPAND should be very useful for patients with XP cutaneous disease, but not for cases of XP neurological disease. As participants, XP subjects with optic or hearing problems should be excluded.
---

REVIEWER	Philip Chilibeck University of Saskatchewan
REVIEW RETURNED	07-Mar-2019

GENERAL COMMENTS	This is a two-armed parallel groups RCT with 24 patients randomized to intervention or delayed intervention. The primary outcome is daily UVR dose to the face measured over 21 days. I have provided mainly a statistical review below. The method used for randomization should be outlined. Outline how missing data will be handled. There is a good chance there will be missing data points during the 21 days of data collection. The sample size is based on the ability to detect an average daily UVR D-to-F difference of 0.10 SED between groups. It is indicated this is clinically meaningful; however, a reference is needed to back up this statement of clinical utility. Please provide clarity for how the primary outcome is being assessed. This seems to require a simple analysis, but the analysis outlined seems complicated. Can this be assessed with a between-groups analysis with values averaged over 21 days? Can the delayed intervention group be assessed with a repeated measures analysis between the first (control) and second (intervention) phase? Will adjustment be made for the multiple secondary outcomes to prevent type I error? Should a CONSORT checklist be included (at least for items up to the reporting of results)...or are the same items covered with the SPIRIT checklist?
--

VERSION 1 – AUTHOR RESPONSE

The following section addresses all the reviewers' comments.

Reviewer 1: Kenneth Kraemer and Deborah Tamura

1. Methodology

a. This is a very select sample and small size (24 patients); randomization will make the intervention sample size even smaller; it may limit eventual generalization to a larger population.

We accept that running a trial in an extremely rare disease is challenging, which is why such conditions have historically been neglected by funding bodies and researchers¹. Our funders, the NIHR, demanded that we adapt standard quantitative methodologies rather than abandon the gold standard aspects of trial design, (i.e., randomisation). On the understanding that it is better to obtain randomised evidence of low power than uncontrolled treatment estimates, randomisation was deemed essential by the research team and the NIHR. The power calculation showed that even with 24 participants we could detect a difference in mean daily dose to the face of 0.10SED, $d=0.73$ (large effect). Furthermore, the study design incorporated a delayed intervention waiting list control, where the control participants also receive the intervention (2019). This allowed us to both compare randomised outcomes both between ($n=12$ vs $n=12$) and within groups “pre-post design” ($n=24$) maximising the information extracted from the small sample. The design was proposed by Dr Adrian Mander, Director of MRC Biostatistics, University of Cambridge who has specialist statistical expertise in running small N RCTS. We consider the findings to be immediately generalizable to international populations of adults with XP and be relevant to other conditions requiring photoprotection (e.g., Systemic Lupus Erythematosus) and healthy populations. XP is rare but the behaviour we wish to change is not. If the intervention is efficacious, we aim to replicate findings in other disease populations.

b. Some XP patients (for example those in complementation groups A or D) are much more photosensitive than others (for example XP complementation groups C or XP variant). They indicate that the patients were stratified by burning type. A reference should be provided as to “burning type”. This stratification will further reduce the sample sizes. Most patients who have extreme burning type will probably practice better sun protection than those who do not have extreme burning because they receive immediate negative feedback by being burned.

Thank you for highlighting this omission. We have added details of the Sunburn Severity Score² explaining how we defined “burning type”.

‘Participants will be randomised, using an equal allocation ratio, to receive XPAND immediately (2018) or to the delayed group stratified by burning type to attempt to balance those with an extreme (i.e., scoring between 1-3 on the sunburn severity score²) versus normal burning response.’

c. Are you taking into account whether the XP patient had a skin cancer removed in the past? This might be a good motivator toward better sun protection.

Due to the limited sample size it is not possible to stratify by multiple factors. Also, while we acknowledge that experience of skin cancer might motivate some patients to change their protection we do not expect the intervention to have different effects in those with, versus without, a history of cancer. Furthermore, in our international survey of >150 XP patients (in submission), we found no relationship between skin cancer and level of photoprotection.

d. Exclusion criteria include “Diagnosed with cognitive impairment”. This is too vague and should be made more specific. For example, some XP patients with mild cognitive impairment may be more amenable to the education than some XP patients with no impairment.

We concede that the exclusion criteria is broad but retain it for two reasons: Firstly, patients at the milder end of the severity continuum might be able to participate at the start of the trial but their condition could deteriorate, impacting the efficacy of the intervention especially if they were randomised to the delayed intervention group. Secondly, as we are unable to assess the impact of

degree of severity as a moderating factor (due to the small sample size) it is better to control by exclusion.

We have improved our description of the exclusion on page 8.

'Diagnosed with cognitive impairment (XP or non-XP related) due to potential impact on the efficacy of the intervention and on the participants' experiences of taking part.'

e. Has it been established that the amount of UV reaching the dosimeter on a participant's wrist modified by the patient's report is actually related to the amount of UV reaching the face? Could the reliability of the reports vary from one patient to another? I do not see that clearly stated in this paper, although some older papers are cited. The readings from this dosimeter are closely associated with the effectiveness of the intervention and the primary study outcome. The comparability of the readings among different individuals should be demonstrated.

The technique of measuring the dose of UVR reaching the face (D-TO-F) is novel. Previous studies have focused on recording UVR exposure, validating self-reported UVR exposure by comparing with objective methods and seeing how these are influenced by different environments and activities³. No study has attempted to fully account for photoprotection by combining it "in the moment" with UVR exposure. To date there is no straightforward way to validate the paradigm as we are not aware of the existence of a comparable measure of UVR at the face. However, the following supports its validity:

-The measurement of UVR at the wrist is valid and reliable ^{4, 5, 6}

-Our previous observational study ⁷ measured D-TO-F in a sample of healthy adults. We hypothesised that the D-TO-F should be higher in this group. As expected the average daily dose was [0.51 SED, 0.01-0.48] compared to [0.13 SED, 0.01-0.48]* for the XP sample.

-To boost reliability and adherence to the protocol, the UVR photoprotection diary was designed to be simple and quick to complete. It uses a grid format which has been shown to be a reliable measure of activities in other contexts ⁸. The day is split into fifteen minute blocks and participants draw a line to show the time they have been outdoors and the type of face protection used during the day. It is a significant advance on previous studies, which assessed protection using a questionnaire at a single time-point, as it is less reliant on memory. Although we expect some individuals to make errors in their recording, we do not anticipate they will be systematic and have equal probability of occurring in the control and intervention groups. In addition, each participant receives training on how to accurately complete the diary.

We acknowledge that due to its originality further validation is required. As it was devised as a surrogate clinical outcome, we would like to monitor the number of skin cancers longitudinally and predict that they will be positively correlated with D-TO-F. It is not possible to do this in the time-frame of the current study.

f. The Photoprotection Self Efficacy Questionnaire and The Brief Photoprotection Adherence Questionnaire are interesting tools that could be used in other clinics and for other photosensitive conditions. Since the tools haven't been validated, the participants' responses may not adequately answer the research questions. It may be worth doing separate validation of these tools for use in the future.

We agree that the value of these questionnaires would be strengthened by future validation. We are particularly interested in the Photoprotection Self Efficacy Questionnaire (PhotoSEQ) as it was designed according to the guidelines for measurement of self-efficacy⁹. Furthermore it provides information about the barriers to protecting in different contexts, which can be used at the group and individual level. In the future we plan to validate it in different populations that need to photoprotect.

g. How much time will be required for each patient to fill out all of the questions required? Is this time reasonable? Is it reasonable to expect each patient to carry out so many daily observations? Adults who have demanding daily activities (such as jobs) may not be as reliable or able to fill out as much of the materials as others who do not have these responsibilities. On the other hand, repeated focusing on sun protection by thinking about filling out the forms may reinforce the message.

We acknowledge that we are asking participants to complete a challenging protocol. However, our previous research 7 had given us confidence that it is doable. Firstly, the short versions of all questionnaires have been selected where available. Secondly, the UVR diary takes 2-3 minutes to complete each day. Participants in our previous study, reported that it was not onerous which was supported by the fact that 38/47 participants produced a total of 775 useable days. In addition, we have worked closely with our Patient and Public Involvement panel who have agreed that the burden of participation is appropriate and unlikely to contribute to attrition.

2. Although the clinic personal have been 'blinded' to the randomization of patients, there may be 'bleed over' of education, support and guidance for the patients who are perceived as not adequately photo-protecting. Most health care personal who work with XP patients, will try to be proactive with all patients who are not photo-protecting.

It is difficult to avoid "bleed over" in a small team working with a rare disease. We are clear with participants not to divulge to which group they had been allocated and we asked staff to continue care as usual. As long as staff do not know which group had received the intervention, it is anticipated it is equally likely that people in both the control and intervention groups would receive opportunistic encouragement to protect, so this would not adversely affect the trial outcome.

3. Please remove references to being "first" or "nobody has attempted". This is scientifically irrelevant and cannot be proven.

We wished to highlight the novelty of the intervention and measurement protocol, but concede we need to acknowledge that this statement is to the best of our knowledge. We have removed the phrase "nobody has attempted" from page 6 and amended the statements on the abstract, pages 4 and 22 as follows:

'Introduction: Poor adherence to photoprotection for people with Xeroderma Pigmentosum (XP) can be life-threatening. A randomised controlled trial (RCT) is being conducted to test the efficacy of a personalised adherence intervention (XPAND) to reduce the level of...(page 2)

'To the best of our knowledge, this is the first RCT to evaluate an adherence intervention designed to improve photoprotection in people diagnosed with XP' (Page 3)

'We aimed to systematically develop an adherence intervention to improve photoprotection in XP. There is growing support for interventions focused on...' (page 5)

'To the best of our knowledge this is the first RCT to test a behaviour change intervention to improve adherence to photoprotection in adults with XP' (page 22)

4. Please reconcile verb tenses in lines 52 to 60 on page 7 and 3 to 11 on page eight. One area is future tense and the other past tense.

Thank you for spotting these inconsistencies. The sentence at the top of page 8 is now written in the future tense.

'Screening and recruitment will take place between February-March 2018'.

Reviewer 2 Eszter Baltas

No comments to address

Reviewer 3 Shinich Moriwaki

XPAND should be very useful for patients with XP cutaneous disease, but not for cases of XP neurological disease. As participants, XP subjects with optic or hearing problems should be excluded.

As noted, we have not designed XPAND to support patients with neurological disease related to XP. We do not consider it appropriate to alter the exclusion criteria for the following reasons:

- Hearing deficits tend to occur alongside cognitive impairment and these patients are likely to already be excluded.
- Ocular damage can be reduced by a photoprotective behaviour, (i.e., wearing UVR protected glasses) which is being targeted by XPAND, and experiencing skin and eye damage are not mutually exclusive.

Reviewer 4 Philip Chilibeck

1. The method used for randomization should be outlined.

We have clarified the description of the randomisation on page 9:

"Participants will be randomised, in blocks using an equal allocation ratio, to receive XPAND immediately (2018) or to the delayed group stratified by burning type to attempt to balance those with an extreme (i.e., scoring between 1-3 on the sunburn severity scale, versus normal burning response. Participants who are in the same family will be randomised as a cluster to the same group to avoid contamination. Since all participants will be recruited at the point of randomisation, the trial statistician (SN) will generate a random allocation for all participants together as using fixed block sizes to ensure equal allocation to both groups. The lead researcher (JWa) will randomly assign (coin toss) group 1 and group 2 to be the intervention or control, to which the statistician will be masked. To protect the integrity of the randomisation, participants in the immediate intervention group will be asked not to reveal their allocation to those outside their immediate family. Group allocation will be concealed from the XP clinical team who are not part of the research team (excluding the PI) to avoid inadvertent changes to the standard care of these participants during the trial (e.g., greater/lesser discussion of adherence during routine clinical appointments)."

2. Outline how missing data will be handled. There is a good chance there will be missing data points during the 21 days of data collection.

Thank you for highlighting this omission. We will assume that data for individual days are missing at random as our analytic approach employs full information maximum likelihood estimation, which allows for unbiased estimates under this assumption (i.e. conditional on variables related to the missing data mechanism being included in the model). We will run sensitivity analyses to explore the influence of missing data on the effect estimates so as to allow for a consideration of the likely plausible range for the treatment effect under a number of assumptions. For average daily assessments, where these are not available for the full 21 days, this will involve imputing missing values under a range of conservative assumptions using the average daily values of the group to which the individual is assigned using a pattern mixture model approach (e.g., average daily level + 0.0SED to 0.5SED). Scale scores will be calculated using proration to account for missing items within the scale where there is at most one-third of items missing. For example, where scale includes

6 items, at least 4 items must have been completed otherwise the scale score will be set to missing for that individual.

A summary of the above has been added to pages 17 and 18 of the manuscript.

3. The sample size is based on the ability to detect an average daily UVR D-to-F difference of 0.10 SED between groups. It is indicated this is clinically meaningful; however, a reference is needed to back up this statement of clinical utility.

To date we do not know the dose of UVR required for clinically meaningful damage of the XP patient. However, data from our observational study measuring predictors of D-to-F (manuscript in preparation) indicate that a reduction of 0.10 SED would require a substantial improvement in photoprotection, considering the range was 0.01-0.48 SEDS*.

4. Please provide clarity for how the primary outcome is being assessed. This seems to require a simple analysis, but the analysis outlined seems complicated. Can this be assessed with a between-groups analysis with values averaged over 21 days? Can the delayed intervention group be assessed with a repeated measures analysis between the first (control) and second (intervention) phase? To clarify, the data from the dosimeter and photoprotection diary will be combined to give a daily estimate of the UVR dose to the face. Across the two three week periods of assessment we will have 42 daily facial UVR dose estimates, which form the outcome variable for the linear mixed model analysis. The marginal mean estimates from these models are interpreted as the average daily dose to the face. Including dummy variables for group and assessment period, as well as their interaction term, allows for the estimate of the average daily UVR D-to F between groups (i.e. treatment effect) for each assessment period.

5. Will adjustment be made for the multiple secondary outcomes to prevent type I error?

We decided that there will be no adjustment for multiple comparisons because the risk of a type II error, in this context, is greater than type I due to the small sample size. However, as a result hypotheses tests will need to be interpreted cautiously.

6. Should a CONSORT checklist be included (at least for items up to the reporting of results)...or are the same items covered with the SPIRIT checklist?

As advised by the editorial team, this comment is disregarded.

VERSION 2 – REVIEW

REVIEWER	Philip Chilibeck University of Saskatchewan, Canada
REVIEW RETURNED	01-May-2019

GENERAL COMMENTS	The authors have responded to my previous comment about how randomization will be performed. I have a couple other comments for this section: - What is the block size? - It is stated the lead investigator will randomly assign group 1 and 2 to intervention and control by a coin toss. I would suggest a
---

	different technique (i.e. random number generator - there are many of these you can find online) and a method (i.e. randomization to blocks) to ensure you don't have an unequal number in intervention and control groups. - minor comment: You need to close the bracket "those with an extreme (i.e. scoring...)
--	--

VERSION 2 – AUTHOR RESPONSE

Reviewer: 4

Reviewer Name: Philip Chilibeck

Institution and Country: University of Saskatchewan, Canada

1. The authors have responded to my previous comment about how randomization will be performed. I have a couple other comments for this section:

What is the block size? It is stated the lead investigator will randomly assign group 1 and 2 to intervention and control by a coin toss. I would suggest a different technique (i.e. random number generator - there are many of these you can find online) and a method (i.e. randomization to blocks) to ensure you don't have an unequal number in intervention and control groups.

-The statistician will randomise in blocks of 4 according to the stratification variable of burning response using a computer programme to generate the random allocation sequence. As part of this process, to ensure masking of group allocation, the groups are referred to as A and B. Separately, the lead researcher randomly assigns group A and group B to be the intervention or control. We acknowledge that this is confusing and have changed the description in the manuscript on page 9 as follows:

'Since all participants will be recruited at the point of randomisation, the trial statistician (SN) will generate a random allocation sequence for all participants together, using a computer programme with fixed block sizes of 4, to ensure equal allocation to both groups.'

2. minor comment: You need to close the bracket "those with an extreme (i.e. scoring...)

Thank you for spotting this and we have added the bracket.

A reference has been updated as it has now been published (Morgan et al., 2019) and doi numbers have been added to the reference list.